# A smoothed particle hydrodynamics study of the collapse for a cylindrical cavity

Andrea Albano◍✉*, Alessio Alexiadis✉*

School of Chemical Engineering, University of Birmingham, Birmingham, United Kingdom

✉ These authors contributed equally to this work.
* AXA1220@student.bham.ac.uk (AA); Alexiadis@bham.ac.uk (AA)

**Data Availability Statement:** All relevant data are within the manuscript and its Supporting Information files.

**Funding:** This work was supported by the US office of Naval Research Global (ONRG) under NICOP Grant N62909-17-1-2051.

## Abstract

In this study, we propose a mesh-free (particle-based) Smoothed Particle Hydrodynamics model for simulating a Rayleigh collapse. Both empty and gas cavities are investigates and the role of heat diffusion is also accounted for. The system behaves very differently according to the ratio between the characteristic time of collapse and the characteristic time of thermal diffusion. This study identifies five different possible behaviours that range from isothermal to adiabatic.

## Introduction

The term "cavitation" describes a phenomenon composed by two distinct phases: first, a vapour cavity, also called vapour bubble or void, develops and rapidly grows in a liquid phase; subsequently, the vapour cavity rapidly collapses generating strong shock waves.

Cavitation causes erosion and it is mostly undesirable in engineering applications such as turbo-machines, propellers, and fuel injectors [1–3]. However, other applications such as ultrasonic cleaning or cataract surgery [4–6] are specifically designed to take advantage of the erosion power of the collapsing bubble.

According to the circumstances, the bubble collapse can follow two distinct, but similar, mechanisms [7] called, respectively, Rayleigh collapse and shock-induced collapse. During the Rayleigh collapse, the collapse is driven by the pressure difference between the surrounding liquid and the cavity. In this case, if the pressure field is perfectly isotropic, the bubble maintains a spherical shape during the whole duration of the collapse. Shock-induced collapse is caused by the passages of a shock-wave through the bubble. In this case, the spherical shape is not preserved and the bubble folds in the shock direction.

Our current understanding of cavitation is based on three different approaches: (i) theoretical investigations, (ii) experiments and (iii) computer simulations.

The first analytical study of an empty cavity surrounded by an incompressible fluid at given pressure was carried out by W. H. Besant (1859) [8], who obtained an integral expression for determining the time required for the cavity to collapse due to the effect of a constant external pressure. Sixty years later, Lord Rayleigh [9] was able to integrate this equation determining that, during the collapse, the pressure of the liquid near the boundary exceeds the pressure of surrounding liquid. Plesset [10] introduced the effect of surface tension and viscosity obtaining

**Competing interests:** The authors have declared that no competing interests exist.

the well-known Rayleigh-Plesset equation that describes the dynamics of a spherical bubble in an infinite body of incompressible fluid. Later, other studies included thermal effect and liquid compressibility [11–14]. Theoretical investigation of the isotropic collapse has continued up to the present day and, recently, Kudryashov & Sinelshchikov [15] found a closed form general solution of the Rayleigh equation for both empty and gas-filled spherical bubbles. The same authors also found an analytical solution of the Rayleigh equation where the surface tension is account for [16].

Experimentally, the study of a collapsing bubble has been a challenge due to difficulty of generating a perfectly spherical bubble, and practical difficulties of measuring relevant data during the short duration of the collapse. The first issue was solved with laser produced cavitation bubbles (eg. [17–19]). This technique, coupled with High-speed photography, increased in particular our understanding of the dynamics of a collapsing bubble in non-isotropic conditions (e.g. near a solid surface) highlighting the role of the so-called jet formation in cavitation erosion. However, the second issue remains an open challenge. In fact, theoretical studies (eg. [20, 21]) calculated temperatures inside the collapsing bubbles to be between 6700 K and 8800 K and pressures up to 848 bar. These peak values, however, occur only for very small intervals of time ($\approx 2\mu$s) and, up to now, the short timescale has prevented accurate experimental analysis of the phenomenon.

The use of computer simulations for investigate cavitation is more recent. Computer simulation can perform "numerical experiments" that, contrary to actual experiments, are not limited by short time-scales and small bubble sizes. During the years, a variety of simulations methods have been used for simulating the collapse of a bubble both near and away from a solid surface: Plesset-Champan used the particle-in-cell method [22], Blake used the boundary integral method [23], Klaseboer [24] used the boundary element method, while Johnsen [7] a high-order accurate shock- and interface-capturing scheme.

All these studies are based on mesh-based computational methods. Meshfree methods are generally considered easier to implement for highly deformable interfaces [25] but, surprisingly, only few articles have simulated cavitation with meshfree methods. One of the few exceptions is Joshi et al. [26, 27] that took advantage of the meshfree nature of Smoothed Particle hydrodynamics (SPH) to develop an axisymmetric model simulating not only the collapse of the cavity, but also the effect of the shock waves on a nearby solid surface (e.g. deformation, erosion). However, the empty cavity used in their model prevents thermal analysis. Albano & Alexiadis [28] developed a SPH model for shock wave interacting with a discrete gas inhomogeneity, this phenomenon share similar physics to the shock induced collapse [29].

This study proposes the first SPH model simulating a Rayleigh collapse of a cavity filled with non-condensable gas induced by abruptly change in pressure. Moreover, by implementing the diffusive heat transfer mechanism, both adiabatic and heat diffusive collapse are simulated. The aim is to investigate the role of heat diffusion in the pressure and temperature development.

The role of heat transfer in reducing the peak temperature of the collapse is known since the '80s [21]. Nevertheless, the diffusion mechanism is often neglected in modelling work and the collapse is assumed adiabatic without justification [30, 31].

## Smoothed particle hydrodynamics

Originally, Gingold and Monaghan [32] and Lucy [33] developed Smoothed-Particle Hydrodynamics (SPH) as a mesh-free particle method for solving astrophysical problems. However, earliest applications also focused on solving fluid dynamics problems [34–36]. In fact, SPH has major advantages in simulating free surface flows and large deformations due to its Lagrangian

nature [37, 38]. The method has been validated for wide range of applications such as explosion [39], underwater explosion [40], shock waves [28, 41, 42], high (or hyper) velocity impact [43], water/soil-suspension flows [44], free surface flows [45, 46], nano-fluid flows [47], thermo-fluid application [48]. Moreover, SPH is also a component of the Discrete multi-physics simulations [49–53]

SPH bases its discrete approximation of a continuum medium on the expression

$$f(\mathbf{r}) \approx \iiint f(\mathbf{r}')W(\mathbf{r} - \mathbf{r}', h)d\mathbf{r}', \tag{1}$$

where $f(\mathbf{r})$ is any continuum function depending of the three-dimensional position vector $\mathbf{r}$, while $W$ is the smoothing function or kernel. The kernel function $W$ defines the extension of the support domain, the consistency, and accuracy of the particle approximation [25]. When the computational domain is divided in computational particles with their own mass, $m = \rho dr$, it is possible to rewrite Eq 1 in particle form

$$f(\mathbf{r}) \approx \sum \frac{m_i}{\rho_i} f(\mathbf{r}_i)W(\mathbf{r} - \mathbf{r_i}, h), \tag{2}$$

where $m_i$, $\rho_i$ and $\mathbf{r}_i$ are mass, density and position of the $i^{th}$ particle. Within the SPH framework, it is possible to use Eq 2 to discretise a set of equations such as the continuity equation

$$\frac{d\rho}{dt} = -\rho \nabla \cdot \mathbf{v}, \tag{3}$$

which in particle form becomes

$$\frac{d\rho_i}{dt} = \sum_j m_j \mathbf{v}_{ij} \nabla_j W_{ij}; \tag{4}$$

the momentum equation

$$\frac{d\mathbf{v}}{dt} = -\frac{1}{\rho} \nabla \cdot P, \tag{5}$$

which in particle form becomes

$$m_i \frac{d\mathbf{v}_i}{dt} = \sum_j m_i m_j \left( \frac{P_i}{\rho_i} + \frac{P_i}{\rho_i} + \Pi_{ij} \right) \nabla_j W_{ij}; \tag{6}$$

where $\Pi_{ij}$ is called artificial viscosity and was introduced by Monaghan [41] for simulating shock waves, having the expression

$$\Pi_{ij} = -\beta h \frac{c_i + c_j}{\rho_i + \rho_j} \frac{\mathbf{v}_{ij} \cdot \mathbf{r}_{ij}}{r_{ij}^2 + \epsilon h^2}, \tag{7}$$

where $\beta$ is the dimensionless dissipation factor, $c_i$ and $c_j$ are the speed of sound of particle $i$ and $j$.

The energy conservation equation

$$\frac{de}{dt} = -\frac{1}{\rho} \left( P(\nabla \cdot \mathbf{v}) + \tau : \nabla \mathbf{v} \right) - \frac{1}{\rho} \nabla \cdot (-k\nabla T), \tag{8}$$

which in particle form becomes

$$m_i \frac{de_i}{dt} = \frac{1}{2} \sum_j m_i m_j \left( \frac{P_i}{\rho_i} + \frac{P_i}{\rho_i} + \Pi_{ij} \right) : \mathbf{v}_{ij} \nabla_j W_{ij} - \sum_j \frac{m_i m_j}{\rho_i \rho_j} \frac{(\kappa_i + \kappa_j)(T_i - T_j)}{r_{ij}^2} \mathbf{r}_{ij} \cdot \nabla_j W_{ij}, \quad (9)$$

where $\kappa$ is the thermal conductivity. The first term of the right side of Eq 9 is the particle form of the sum of the reversible rate of the internal energy increase by compression and the irreversible rate of internal energy increase by viscous dissipation; the second term is the particle form of the rate of internal energy increment by heat conduction following Fourier's Law. The thermal conductivity is related to the thermal diffusivity, $\alpha$, by the relationship

$$\alpha = \frac{\kappa}{\rho c_p}, \quad (10)$$

where $c_p$ is the specific heat capacity.

## Kernels function

In this work, two kernels (Lucy Kernel function and quintic spline) are used and their effect on the accuracy of the results compared. The Lucy kernel function [33]

$$W(q, h) = \begin{cases} \frac{1}{s} [1 + 3q][1 - q]^3, & q \leq 1 \\ 0, & q > 1, \end{cases} \quad (11)$$

is one of the simplest kernels used in literature. Where $q = |\mathbf{r} - \mathbf{r}'|/h$, $s$ is a parameter used to normalise the kernel function, which, for one, two and three dimensional space is, respectively, $\frac{4h}{5}$, $\frac{\pi h^2}{5}$ and $\frac{16\pi h^3}{105}$. The quintic spline is a piecewise kernel function [54]

$$W(Q, h) = s \begin{cases} (3 - q)^5 - 6(1 - q)^5 + 15(1 - q)^5, & 0 < q < 1 \\ (3 - q)^5 - 6(1 - q)^5, & 1 < q < 2 \\ (3 - q)^5, & 2 < q < 3 \\ 0, & q > 3 \end{cases} \quad (12)$$

where $q = |\mathbf{r} - \mathbf{r}'|/h$ and $s$ for one, two and three dimensional space is, respectively $\frac{1}{120h}$, $\frac{7}{478\pi h^2}$ and $\frac{3}{359\pi h^3}$. In the quintic kernel is normally more accurate, but at the expenses of higher computational costs because it requires a neighbor list three times larger than the Lucy kernel [25].

## Model

**Problem description.** In the Rayleigh collapse, the driver force is the pressure difference between the pressure in the liquid, $p_\infty = P_L$, and the pressure in the cavity, $p_b$.

Two different scenarios are analysed: empty cavity collapse, where the cavity is void, with $p_b = 0$, surrounded by a liquid phase, and vapour cavity collapse, where the cavity is filled of a non condensable gas with an initial pressure equal to the vapour pressure of water at the temperature $T_0$, $p_b = p_{sat}(T_0)$ and density equal to the density of an ideal gas at that pressure and temperature, $\rho_b(p_b, T_0)$.

The liquid phase is water with $\rho_L = 1000$ kg/m$^3$, $P_L = 5$ MPa and $T_0 = 300$ K. The liquid pressure of 5 MPa has been chosen as it is commonly reached in various hydraulic applications [55]. Given the liquid temperature, the pressure in the bubble is $p_b = 3.55$ kPa with a density of

$\rho_b(p_b, T_0) = 2.7 \cdot 10^{-3}$ kg/m$^3$. At the short timescale considered, water is compressible. The equation of state for compressible water is discussed later on.

In rapid collapse, the water vapour is considered trapped within the liquid, assuming zero mass transport across the interface. This simplification is justified because mass transport mechanisms across the interface and non-equilibrium condensation require higher timescales to play an effective role in the collapse phase [56]. The short timescale also justifies neglecting the surface tension in modelling the collapse.

The short timescale of the phenomenon may suggest an adiabatic collapse [20, 57]. However, especially in the last stage of the collapse, the high temperature gradient between gas and liquid phases could introduce a non-negligible heat transfer between the two phases [21, 58]. In this study, both scenarios (e.g. adiabatic and non-adiabatic collapse) are investigated.

**SPH model.** The axisymmetric water domain is shown in Fig 1. The domain is dived in three concentric regions, delimited by three different radiuses, where different types of computational particles are used.

Cavity ($r < R_0$): inside this region particles are removed (in the case of empty cavity) or modelled as non-condensable gas following a gas phase equation of state (EOS) in the case of vapour collapse. In the rest of the paper, particles inside the cavity (when present) will be referred as particle Type 1.

Liquid ($R_0 < r < R_S$): inside this region particles are modelled as compressible fluid following a liquid EOS. Particles inside the liquid will be referred as particle Type 2.

Shell ($r > R_S$): inside this region particles are modelled as fluid with a fixed position and density to represent the boundary conditions of the system and maintain a fixed pressure at the boundaries. Particles inside the shell will be referred as particle Type 3. We also run several simulations with cubic control volumes and periodic conditions that account only for Type 1 and Type 2 particles. The results do not change and, therefore, we prefer the system in Fig 1 that overall requires less computational particles.

To avoid compenetration between gas and liquid particles during the gas cavity collapse, we employed a penalty force, similar to the one used by Liu et al. [40] between these types of particles.

$$f_p = \begin{cases} -C \dfrac{\gamma}{r_{ij}} \left( \dfrac{\sigma}{r_{ij}} \right)^{\gamma}, & r_{ij} \leq \sigma \\[2em] 0, & r_{ij} > \sigma, \end{cases} \tag{13}$$

with $C = 10^{-4}$, $\gamma = 9$ and $\sigma$ equal to the initial particle spacing.

The initial radius of the cavity is $R_0 = 100\mu$m (typical radius of a collapsing cavity [26, 59]). The ratio $R_C/R_0 = 30$ is used as a compromise between computational cost and accuracy. Different $R_S$ has been tested, as explained in the Hydrodynamic section.

Different resolutions (i.e. total number of computational particles) have been tested ($5.79 \cdot 10^5$, $1.30 \cdot 10^6$ and $2.66 \cdot 10^6$); $N = 1.06 \cdot 10^6$ was chosen as best compromise between accuracy and computational speed (more details in the Hydrodynamic section.

**Equation of state.** To solve the set of Eqs 3–8, an EOS that links pressure $P$ and density $\rho$ is required. Each phase requires a different EOS: in this work multiple EOS are used and compared.

*Liquid EOS*. For liquids, we used and compared two EOS: the Tait and the Mie-Gruneisen EOS.

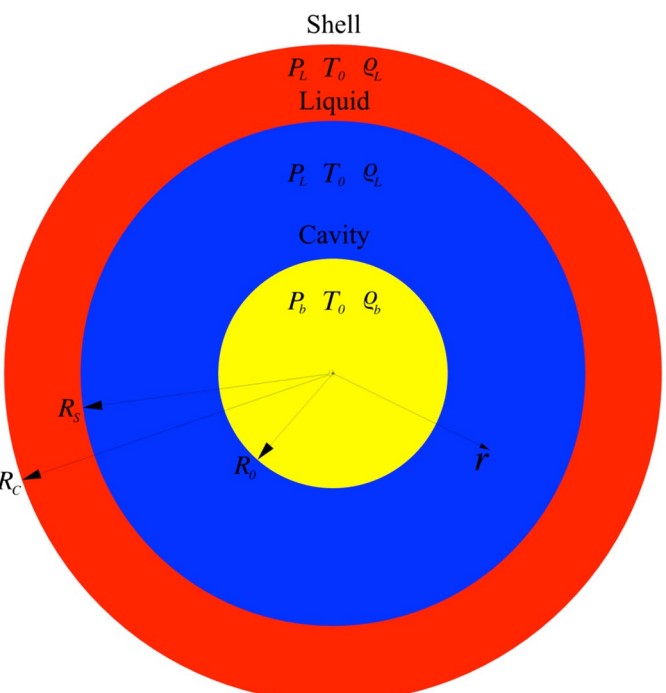

**Fig 1. Geometry of the simulation box.**

The Tait equation is probably the most used EOS in SPH to model water

$$P(\rho) = \frac{c_0^2 \rho_0}{7}\left(\left(\frac{\rho}{\rho_0}\right)^7 - 1\right), \tag{14}$$

where $c_0$ is speed of sound of the liquid and $\rho_0$ is the reference density. The Tait EOS takes in account the compressibility of the liquid. Is possible to regulate the compressibility by selecting the appropriate sound of speed [60] in Eq 14.

For simulating underwater explosion Liu et all [40] used the Mie-Gruneisen EOS [61] to model the water as a compressible fluid having different expressions for compression and expansion state. Shin et al. [62] derived a polynomial expression for both compression and expansion states: for compression state,

$$P(\rho, e) = a_1 \mu + a_2 \mu^2 + a + 3\mu^3 + (b_0 + b_1 \mu + b_1 \mu^2)\rho_0 e, \tag{15}$$

while for expansion state,

$$P(\rho, e) = a_1 \mu + (b_0 + b_1 \mu)\rho_0 e, \tag{16}$$

Where $\mu = \rho/\rho_0 - 1$ and $e$ is the specific internal energy. The coefficients are $a_1 = 2.19 \cdot 10^9$ N/m$^2$, $a_2 = 9.224 \cdot 10^9$ N/m$^2$, $a_3 = 8.767 \cdot 10^9$ N/m$^2$, $b_0 = 0.4934$ and $b_1 = 1.3937$ evaluated for water with $\rho_0 = 1000$ kg/m$^3$ and $c_0 = 1480$ m/s.

*Vapour EOS.* The vapour phase in the cavity is modelled as a non-condensable gas. In our simulations, we used and compared two EOS: the ideal gas EOS and the NASG EOS.

The ideal gas EOS, for the pressure, is given by

$$P(\rho, e) = (\gamma - 1)\rho e, \tag{17}$$

temperature

$$T(e) = M_m \frac{(\gamma - 1)e}{R},$$

(18)

where $\gamma = c_p/c_v$ is the capacity heat ratio, $M_m$ the molar mass of the gas and $R$ is the ideal gas constant. The NASG EOS, which is a multiphase EOS is discussed in the next section.

*Multiphase EOS.* Le Métayer & Saurel [63] combined the "Noble-Abel" and the "Stiffened-Gas" EOS proposing a EOS called Noble-Abel Stiffened-Gas (NASG), suitable for mulfiphase flow. The expression of the EOS does not change with the phase considered, and, for each phases, is possible to determine both the pressure and temperature as function of density and specific internal energy. Pressure-wise the expression of NASG is

$$P(\rho, e) = (\gamma - 1) \frac{(e - q)}{\left(\dfrac{1}{\rho} - b\right)} - \gamma P_\infty,$$

(19)

and temperature wise

$$T(\rho, e) = \frac{e - q}{C_v} - \left(\frac{1}{\rho} - b\right) \frac{P_\infty}{C_v},$$

(20)

where $P$, $\rho$, $e$, and $q$ are, respectively, the pressure, the density, the specific internal energy, and the heat bond of the corresponding phase. $\gamma$, $P_\infty$, $q$, and $b$ are constant coefficients that defines the thermodynamic properties of the fluid. The coefficients for liquid water and steam used in our simulations are given in Table 1.

**Software for simulation, visualisation and post-process.** All the simulation were run with the open source code simulator LAMMPS [64, 65]. Visualisation and data post-processing were generated with the Open Source code OVITO [66].

## Hydrodynamic

**Empty cavity.** Different simulations have been run to assess the quality of results with respect of numerical parameters such as number of computational particles, kernel function and time step. Preliminary simulations have been run using both Lucy (Eq 11) and quintic spline (Eq 12) kernel functions obtaining similar results. Therefore, we chose the Lucy kernel over the over the quintic because it requires less computational cost because accounts for a smaller neighbour list. In all cases smoothing length and the dissipation factor are $h = 1.3 \cdot dL$, where $dL$ is the initial particle spacing, and $\beta = 1$, coherent with literature in shock-wave problems [25, 42].

**Table 1. NASG coefficients for liquid water and steam.**

| Coefficient | Liquid phase | Vapor phase |
|---|---|---|
| $C_p$ [J kg$^{-1}$ K$^{-1}$] | 4285 | 1401 |
| $C_v$ [J kg$^{-1}$ K$^{-1}$] | 3610 | 955 |
| $\gamma$ [-] | 1.19 | 1.47 |
| $P_\infty$ [Pa] | $7028 \cdot 10^5$ | 0 |
| $b$ [m$^3$kg$^{-1}$] | $6.61 \cdot 10^{-4}$ | 0 |
| $q$ [J kg$^{-1}$] | -1177788 | 2077616 |
| $q'$ [J kg$^{-1}$ K$^{-1}$] | 0 | 14317 |

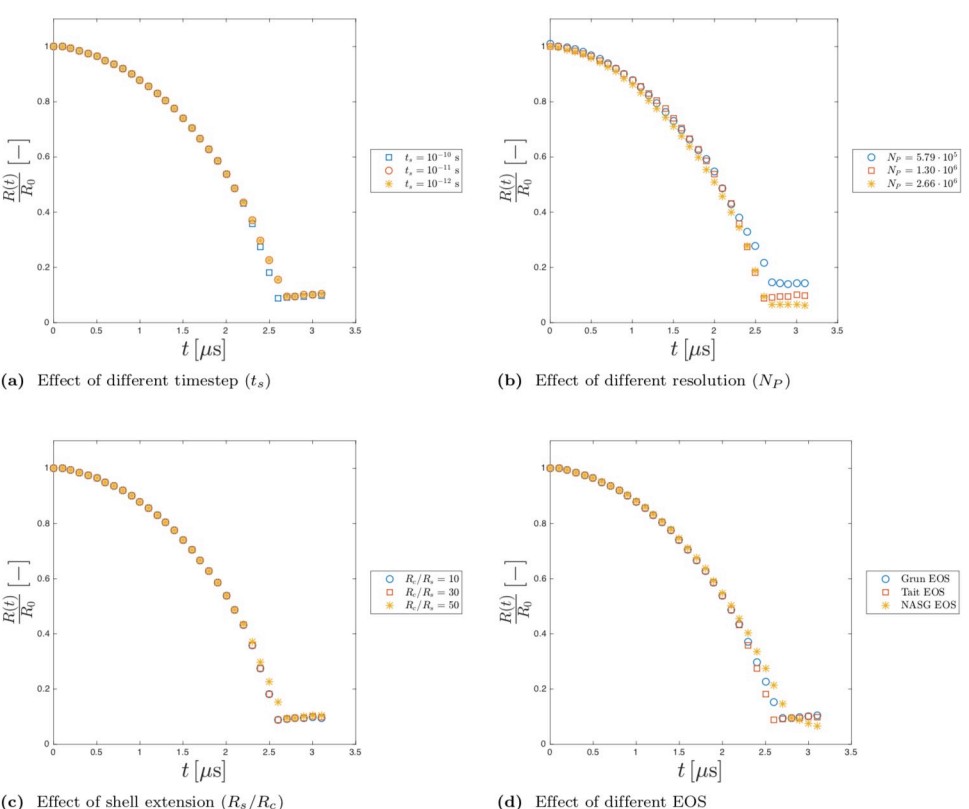

**Fig 2. Effect of different simulation parameters on the Hydrodynamic the SPH model.** A: Effect of different timestep ($t_s$). B: Effect of different resolution ($N_P$). C: Effect of shell extension. $R_s/R_c$ D:Effect of different EOS.

Fig 2 shows the evolution of the dimensionless radius $R(t)/R_0$ of a collapsing cavity. The collapsing time obtatined with the SPH model is around 2.76$\mu$s, which is very close to $t_c$ = 2.70$\mu$s the collapsing time obtained by solving the axisymmetric Rayleigh-Plesset (ARP) equation [67].

Fig 2 summarise the effect of different parameters on the simulation: Fig 2(a) shows the effect of different timestep, the higher timestep value, $t_s$ = $10^{-10}$ s, was chosen according to the CFL criterion. Fig 2(b) shows the effect of different resolution (number or particles). Fig 2(c) shows the effect of the extension of the shell region, expressed with the ratio $R_C/R_S$. Fig 2(d) shows the profile of the collapse obtained with different liquid EOS.

Note that our dimensionless radius does not goes to zero, but it rebounds, like the axisymmetric Raylerigh-Plesset equation, this is explainable with the particle nature of the SPH method: with particle methods, in fact, there is always a small spacing between particles.

Based on the analysis of this section we decided, for an empty cavity collapse, to use the parameters shown in Table 2.

All the parameter listed are chosen as the best compromise between speed an accuracy.

**Table 2. Parameters used for simulating the empty cavity Rayleigh collapse.**

| Kernel | Water EOS | h | $\beta$ | $t_s$ [s] | $N_p$ | $R_s/R_c$ | $c_0$ [m s$^{-1}$] |
|---|---|---|---|---|---|---|---|
| Lucy Kernel | Tait | $1.3 \cdot dL$ | 1 | $10^{-10}$ | $1.30 \cdot 10^6$ | 0.3 | 1484 |

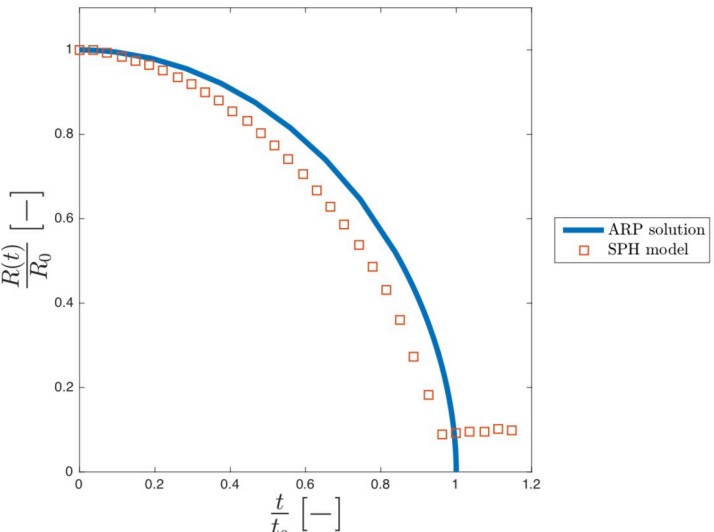

**Fig 3. Effect of different timestep (t$_s$) on the energy trend.**

**Vapour cavity.**   Fig 3 shows the trend of the dimensionless internal energy, $e/e_0$, versus dimensionless time for different timestep.

Where $e_0$ is the initial internal energy of particles. The timestep $t_s = 10^{-12}$ s was chosen since it is the largest timestep where the internal energy decreases after the collapse as required by the physics of the problem. The parameters used are summarised in Table 3.

**Comparison with the axisymmetric Rayleigh-Plesset equation.**   The Hydrodynamic of the model is compared with the solution of the axisymmetric Rayleigh-Plesset (ARP) Eq [7] for both the empty and vapour cavity. Fig 4 shows the dimensionless radius, $R(t)/R_0$, plotted against dimensionless time, $t/t_c$, of our model against the solution of equation ARP for the empty collapse. $R_0$ is the initial radius of the cavity, $t_c = 2.76\mu s$ is the collapsing time obtained with the ARP.

In our model, the cavity collapse slightly faster than the theoretical, leading to $t/t_c \approx 0.98$ instead of 1. This difference is explainable with the compressibility of the liquid [13]. The theoretical model assumes that water is perfectly incompressible, while the Tait EOS in the SPH model accounts for the compressibility of water. At these timescales, the compressibility cannot be neglected, and, from this point of view, our compressible SPH model should be more accurate than the theoretical, fully incompressible model. It is also important to highlight that, in our simulation, the parameter $c_0$ (sound speed in the medium) in the Tait EOS, we use is 1484 m/s, which is the actual speed of sound in water at 25˚C.

According to our calculations, the effect of the compressibility affects the rate of collapse. At the beginning, the compressibility produces a higher collapsing rate because a compressible fluid fills the void in the cavity faster than an incompressible fluid. As the cavity shrinks, however, the curvature of the cavity acts as an arch and the speed of the collapse slows down more than in the more rigid (incompressible) case. Overall, these two effects cancels each other out

**Table 3. Parameters used for simulating the vapour cavity Rayleigh collapse.**

| Kernel | Water EOS | Gas EOS | h | $\beta$ | $t_s[s]$ | $N_p$ | $R_s/R_c$ | $c_0$ [m s$^{-1}$] |
|---|---|---|---|---|---|---|---|---|
| Lucy Kernel | Tait | Ideal gas | $1.3 \cdot dL$ | 1 | $10^{-10}$ | $1.30 \cdot 10^6$ | 0.3 | 1484 |

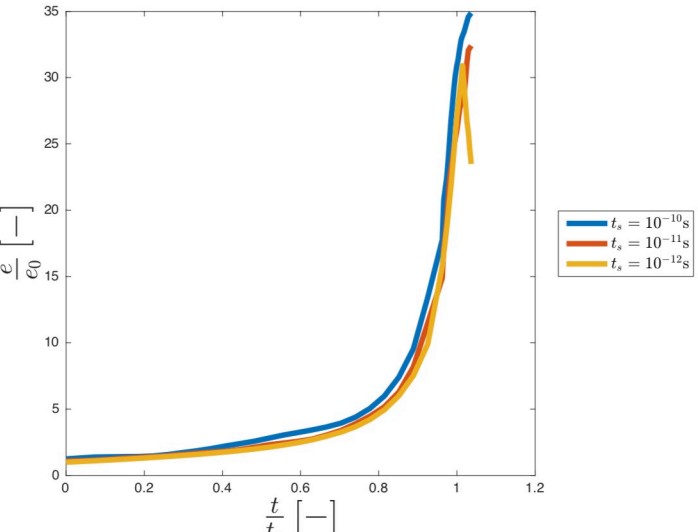

**Fig 4. Dimensionless ratio (R/R$_0$) against dimensionless time (t/t$_c$) for both SPH (square dot) and ARP (continuum blue curve) for the empty cavity collapse.**

and the final collapsing time, is almost identical in the case of the theoretical Rayleigh-Plesset (incompressible) case and the SPH model based on the (compressible) Tait EOS.

Fig 5 shows the dimensionless radius, $R(t)/R_0$, plotted against dimensionless time, $t/t_c$, of our model against the solution of equation ARP for the vapour collapse.

The considerations done for the empty collapse are still valid for the vapour cavity case. However, additional discussion is required for the final phase of the collapse and the rebound phases: unlike the empty cavity (see Fig 6a) when the vapour cavity approaches the final phase of the collapse, the cavity loses the cylindrical symmetry (Fig 6b), and differs from the ARP.

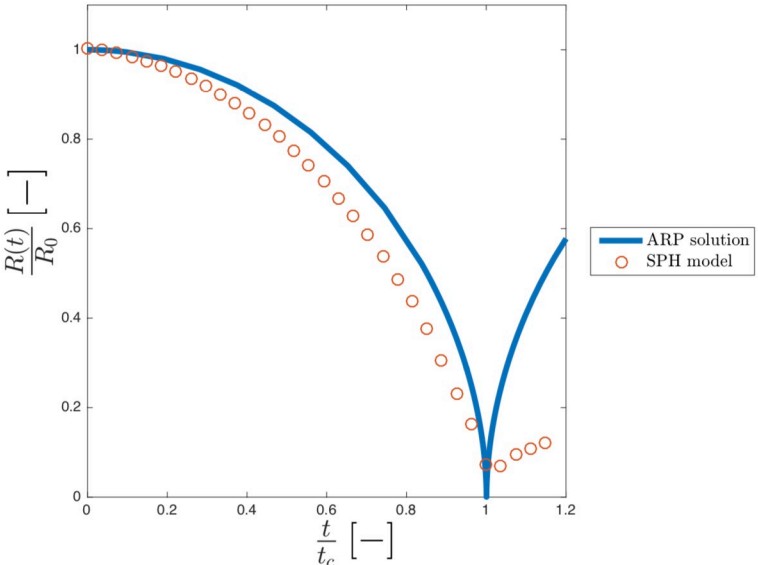

**Fig 5. Dimensionless ratio (R/R$_0$) against dimensionless time (t/t$_c$) for both SPH (square dot) and ARP (continuum blue curve) for the vapour cavity collapse.**

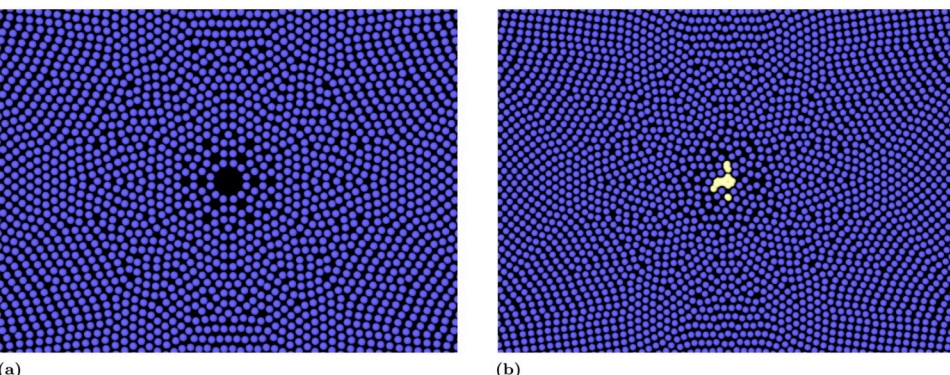

**Fig 6.** Final stage of the collapse for the empty cavity (a) and the vapour cavity (b).

This "artificial" asymmetry in the rebound phase is attributable to low resolution occurring when, during the last stage of the collapse, the size of the caivty is comparable to the size of the smoothing length. However, this work focuses only on the bubble collapse (as usual in computer simulations of cavitation e.g. [7, 26, 59]) and the rebound phase is not considered.

## Results and discussion

### Pressure field

**Pressure field in the liquid (empty cavity).** Initially, at $t = 0$, the pressure is uniform along the domain. As the cavity shrinks, due to the pressure difference between the liquid and the cavity, the liquid starts to fill the cavity. This causes a decrement in pressure in the liquid generating a low-pressure wave that moves through the liquid phase (see Fig 7).

As the collapse proceed, a high-pressure area arises near the cavity border (see Fig 8a and 8b) that abruptly increases reaching the max at the collapse (see Fig 8c and 8d). Locally, the max pressure calculated is around 120 MPa. This value is one order of magnitude lower than the theoretical value calculated by Hickling and Plesset [68] for the 3D collapse, but this difference is consistent with the fact that our model refers to a 2D collapse [67, 69]. This is also reflected in the difference in the collapse time between 2D and 3D [9, 70]

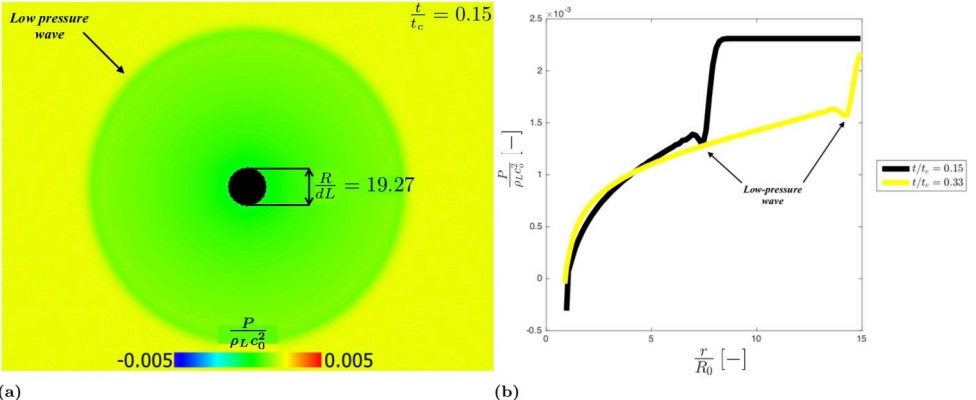

**Fig 7.** Dimensionless pressure field in the liquid phase for $t/t_c = 0.15$ and R/dL = 19.27 (a); Dimensionless pressure spatial trend for $t/t_c = 0.15$ and $t/t_c = 0.33$ (b).

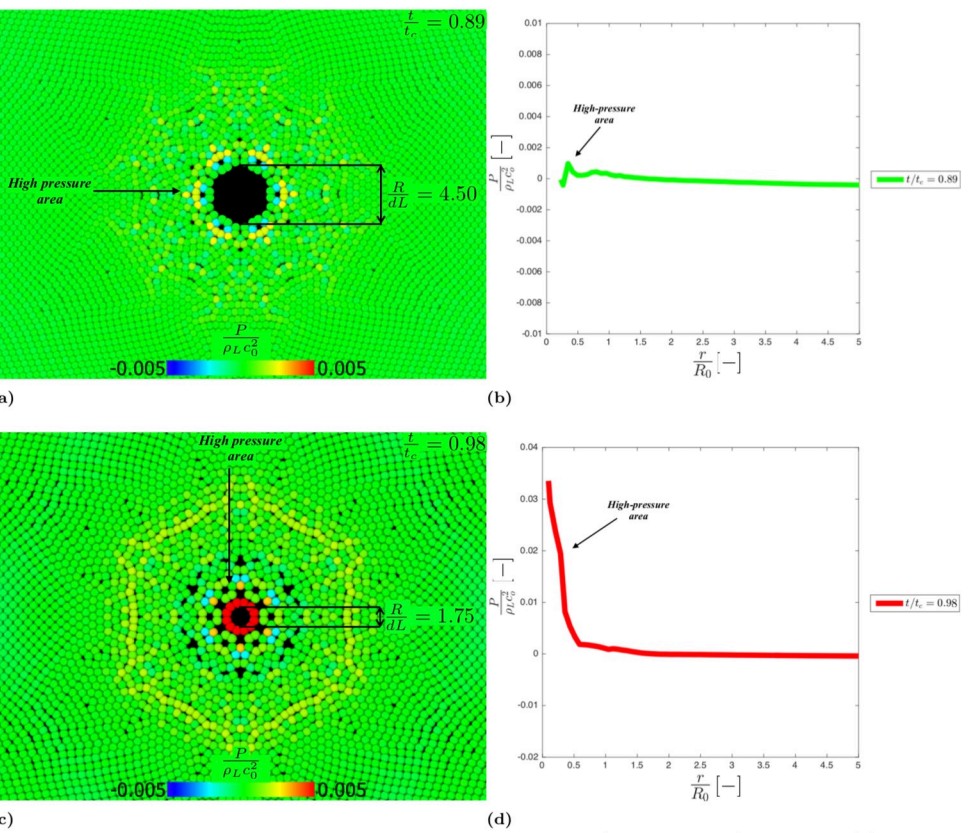

**Fig 8.** Dimensionless pressure field in the liquid phase for $t/t_c = 0.89$, $R/dL = 4.50$ (a) and $t/t_c = 0.98$, $R/dL = 1.75$ (d); Dimensionless pressure spatial trend for $t/t_c = 0.89$ (b) and $t/t_c = 0.98$ (d).

The collapse generates a high-pressure wave (Fig 9 and 9b) that moves away form the cavity (Fig 9c and 9d). There is a theoretical reason for the hexagonal patterns in Figs 8 and 9, which is discussed in the next section.

With the absence of heat diffusivity the presence of the vapour in the cavity does not affect significantly the pressure in the liquid and, therefore, pressure fields for the vapour cavity collapse are not shown here.

**Acoustic diffraction.** As mentioned in the previous section, when the empty cavity reaches the minimum radius, a high-pressure shock wave is generated and propagates in the liquid phase. After the wave bounces back, it loses its spherical symmetry and assumes an unphysical hexagonal symmetry.

This is a numerical artefact and depends on the fact that, below a certain raito $R/dL$, the initial particle resolution is not adequate to correctly describe the cavity shape (the reader can compare Fig 7, where $R/dL = 19.27$, with Fig 8, where $R/dL = 4.50$-$1.75$). The cavity assumes a hexagonal shape (see Fig 8a), which is related with initial hexagonal particle distribution of the model. When the high-pressure shock wave bounces back, therefore, it propagates from a hexagonal cavity rather than a circular one.

This behaviour closely resembles light diffraction from a hexagonal aperture, Fig 10. Light diffraction, in fact, follows specific patterns [71] defined by the shape of the aperture.

Fig 10 shows the similarity between the light intensity pattern generated by diffraction trough a hexagonal opening and the pressure intensity pattern of the wave generated at the

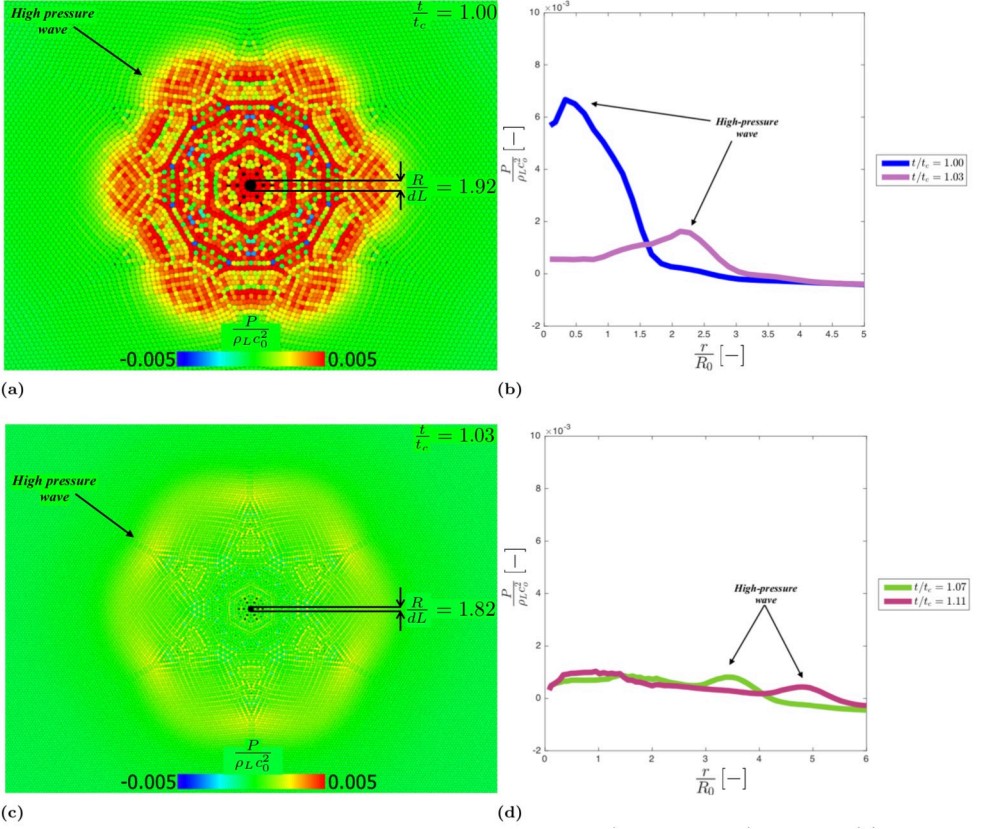

**Fig 9.** Dimensionless pressure field in the liquid phase for $t/t_c = 1.00$, R/dL = 1.92(a) and $t/t_c = 1.03$, R/dL = 1.82 (d); Dimensionless pressure spatial trend for $t/t_c = 1.00$, $t/t_c = 1.03$(b) and $t/t_c = 1.07$, $t/t_c = 1.11$ (d).

collapse of the cavity. The pressure peaks and valleys in Fig 10b (and Fig 9a), therefore, are the results of Fresnel like positive and negative interference of the interfering diffracted waves rather than the result of effect of numerical instability.

This issue, however, only occurs at the end of the collapse, when the size of the cavity is comparable to the smoothing length and does not affect the overall collapsing time.

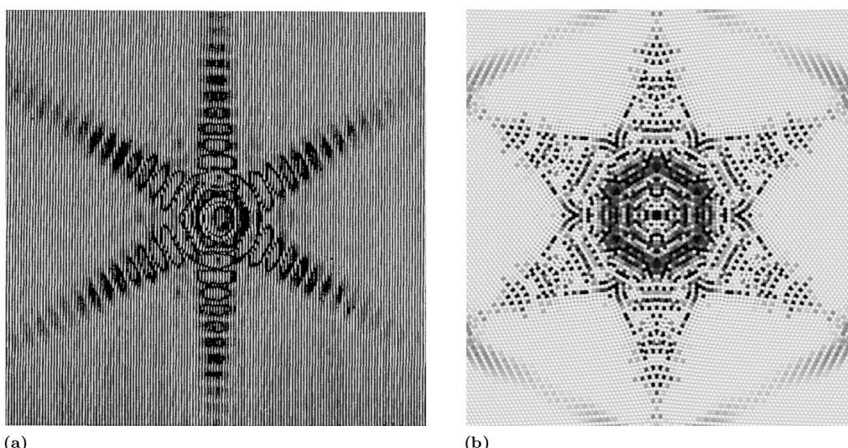

**Fig 10.** Comparison between light intensity for a hexagonal aperture [71] (a) and pressure intensity calculated with our model (b).

## Thermal dffects

During the collapse, the compression of gas in the bubble generates heat. This heat, in turn, can affect the dynamic of the collapsing bubble [14]. When thermal effects are absent, or negligible, the collapse is "inertially controlled" as in the previous Section. When the thermal effects are not negligible, the collapse can be "thermally controlled". In a thermally controlled collapse, the bubble dynamic differs from the inertially controlled because the thermal terms in the ARP equation are not negligible.

Two scenarios are analysed. In the adiabatic collapse, only the first term of Eq 9 is accounted for. In the heat diffusive collapse, both term are enabled to model heat transfer between gas and liquid.

Finally, the temperature peak in the gas cavity is investigated in relation to the ratio between the characteristic time of collapse and the characteristic time of heat transfer.

**Adiabatic collapse.** The average pressure and the temperature in the vapour cavity increase during the collapse (see Fig 11), locally reaching a max $P \approx 40$ MPa and $T \approx 10000$ K. Those values, despite some difference in the operating conditions, are comparable to those measured by Obreschkow et al. [72].

The pressure and temperature field distribute differently in the cavity:

1. In the first stage of the collapse, the interaction between gas and fluid results in a rapid increment of pressure at the cavity interface (Fig 12a) generating a shock wave inside the cavity. Later, because of the combined effect of cavity compression and shock wave propagation, the pressure increases in the centre of bubble (Fig 12b) becoming almost uniform at the collapse.

2. Similarly to the pressure, the temperature increases at the cavity interface during the first stages of the collapse (see Fig 12c). The absence of heat diffusion mostly affects the final stage of the collapse: the internal energy does not diffuse and heat is confined and accumulated at the cavity interface (Fig 12d).

**(Heat) Diffusive collapse.** Our results show that time of the collapse is not significantly affected by the presence of heat transfer in the model. However, the pressure and temperature peak inside the bubble decreases with respect to the adiabatic case, see Fig 13.

Also the pressure and temperature fields inside the cavity change:

1. In the first phase of the collapse, the pressure field in the cavity remains uniform (Fig 14a). Approaching the final stage of the collapse, the pressure is slightly lower at the cavity interface, Fig 14b, because of the presence of the diffusive heat transfer mechanism. In fact, the pressure of an ideal gas is function of both the density and the internal energy, as shown by Eq 17. The presence of a high temperature gradient at the cavity interface greatly reduces the internal energy of the particles in that area.

2. Similarly to the adiabatic case, in the first stage the interaction between gas and the fluid tend to increment the temperature near the cavity interface, see Fig 14c. However, in this case the heat generated at the interface diffuses both towards the centre of the cavity and into the liquid. This leads to higher temperatures at the centre than at the interface (see Fig 14d).

By comparing Fig 14 with Fig 12, it is clear that, despite the total collapsing time is almost the same, the heat exchange mechanism has important consequences on both temperature

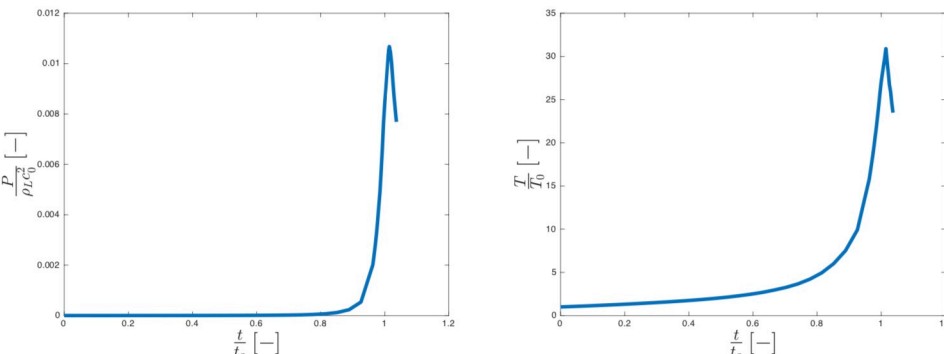

**Fig 11.** Dimensionless pressure trend (a) and dimensionless temperature (b) in the gas phase for the adiabatic collapse.

and pressure in the cavity. Our results, therefore, show that adiabatic conditions, despite being often used in the literature [30, 31, 73–75], are not always realistic. The collapse generates great amount of heat and high temperatures are reached in the cavity. Despite the small timescale of the process, temperature in the cavity rapidly grows and the heat transfer from the cavity interface to the surroundings cannot be neglected.

**Effect of thermal diffusivity on the temperature peak.** In the previous section, we calculated a specific case, where the liquid is water, the gas is water vapour, $\Delta P = P_L - p_G = 5MPa$ and $R_0 = 100\mu m$. In this section, we study how different parameters and initial conditions would affect the temperature rise $T/T_0$ in the cavity. In order to simplify the study, we perform

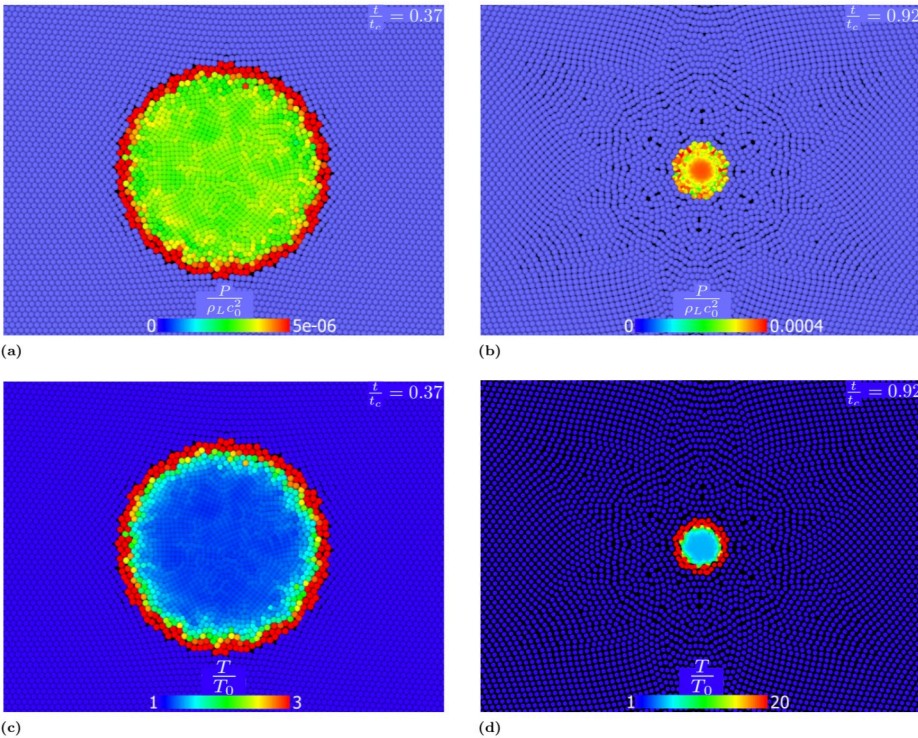

**Fig 12.** Dimensionless pressure field in the cavity for $t/t_c = 0.37$ (a) and $t/t_c = 0.92$ (b) and dimensionless temperature field for $t/t_c = 0.37$ (c) and $t/t_c = 0.92$ (d) for adiabatic collapse.

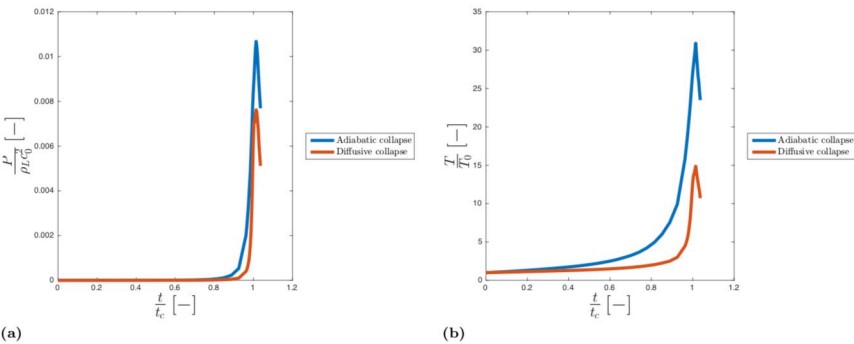

**Fig 13.** Comparison between: Dimensionless pressure trend (a) and dimensionless temperature (b) of the gas phase for adiabatic (blue) and diffusive (red) collapse.

a dimensional analysis of our system to reduce the number of significant parameters. Assuming that the collapse depends on $\Delta P$, $\rho_L$, $\alpha_{L,G} = (\alpha_L + \alpha_G)/2$ and $R_0$, and using the Buckingham $\pi$ theorem, it is possible to determine that the system depends on two fundamental dimensionless groups

$$\Pi_1 = \frac{R}{\alpha_{L,G}} \sqrt{\frac{\Delta P}{\rho_l}}, \tag{21}$$

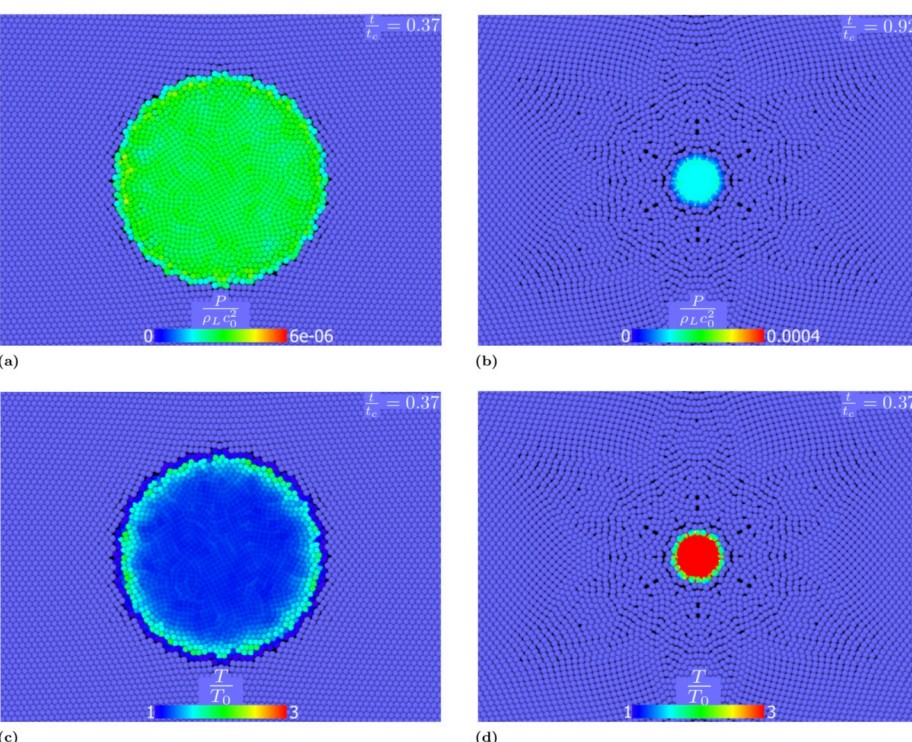

**Fig 14.** Dimensionless pressure field in the cavity for $t/t_c$ = 0.37 (a) and $t/t_c$ = 0.92 (b) and dimensionless temperature field for $t/t_c$ = 0.37 (c) and $t/t_c$ = 0.92 (d) for diffusive collapse. The heat diffusivity in the liquid is $\alpha_L = 1.48 \cdot 10^{-7}$ m$^2$/s (liquid water) in the gas $\alpha_G = 4.09 \cdot 10^{-4}$ m$^2$/s (water vapour).

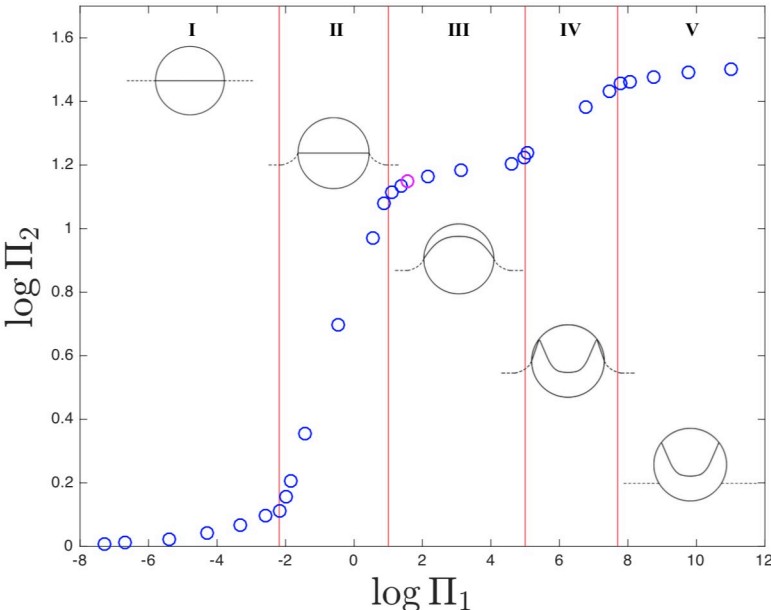

**Fig 15. Temperature peak ($T_{max}/T_0$) as a function of $\Pi_1$.**

$$\Pi_2 = \frac{T}{T_0}. \tag{22}$$

The dimensionless group $\Pi_1$ can also been seen as the ration between the characteristic heat diffusion time scale of the process, $R^2/\alpha_{L,G}$, and the Rayleigh collapsing time, $R\sqrt{\rho_L/\Delta P}$:

$$\Pi_1 = \frac{\tau_d}{t_c} = \frac{R^2}{\alpha_{L,G}} \cdot \frac{1}{R}\sqrt{\frac{\Delta P}{\rho_l}}, \tag{23}$$

When $t_c \ll \tau_d$, the collapse is faster than the characteristic time of heat transfer, the heat generated is trapped in the cavity, and the process can be considered adiabatic. When $t_c \gg \tau_d$, the characteristic time of heat transfer is smaller that the collapsing time and the process can be considered isotherm.

In Fig 15 $T_{max}/T_0$ is plotted against different values of $\Pi_1$.

The "magenta" point represent the collapse analysed in the previous section with $\Pi_1 = 34.76$.

## Conclusion

This work proposes the first SPH model of a collapsing cavity filled with non condensable gas coupled with the heat transfer mechanism.

The aim of the work is to understand the role of diffusive heat transfer during the Rayleigh collapse. This was achieved by introducing the dimensionless group, $\Pi_1$. $\Pi_1$ defined as ratio between the characteristic time of collapse and the characteristic time of thermal diffusion.

In Fig 15 five regions are identified. For each of these regions the temperature field of the gas-liquid system distributes differently:

- *I* region ($0 < \Pi_1 < 6.5 \cdot 10^{-3}$): in this region, both the gas and liquid behave isothermally. As a result of this, the liquid drains all the energy developed by the gas.

- *II* region ($6.5 \cdot 10^{-3} < \Pi_1 < 10$): in this region, the gas behaves isothermally while the liquid shows a temperature profile. The energy rapidly diffuses inside the cavity, flattening the temperature profile in the cavity.

- *III* region ($10 < \Pi_1 < 1 \cdot 10^5$): in this region, neither the gas nor the cavity are adiabatic. This scenario is described in (heat) diffusive collapse section and by Fig 14.

- *IV* region ($1 \cdot 10^5 < \Pi_1 < 5 \cdot 10^7$): in this region, the gas in the cavity behaves adiabatically, but the liquid does not. Therefore, part of the energy generated at the interface is transferred to the liquid phase.

- *V* region ($\Pi_1 > 5 \cdot 10^7$): in this region both the gas in the cavity and the liquid behave adiabatically. All the energy generated by the collapse is trapped in the cavity and the temperature increment is concentrated in the cavity interface (see Fig 12).

In brief, this analysis shows that for $\Pi_1 > 5 \cdot 10^7$ the collapse can be considered adiabatic. At smaller $\Pi_1$, the heat generated at the cavity interface is taken away by the liquid phase, or diffuses in the cavity, homogenising the temperature field. When $\Pi_1 < 6.5 \cdot 10^{-3}$, all the heat generated in the collapse is drained by the liquid and the collapse can be considered isotherm.

This shows that, despite the short timescale, the presence of the heat transfers mechanism leads to a temperature peak drop of around 50% compared to the adiabatic case. This suggest that the adiabatic assumption for the Rayleigh collapse could leads to a not reliable pressure and temperature peak estimation and its use should be properly justified.

## Supporting information

**S1 File. Input file for LAMMPS.** All the results presented in this work were obtained with this input file.
(LMP)

## Author Contributions

**Conceptualization:** Andrea Albano, Alessio Alexiadis.

**Data curation:** Andrea Albano.

**Funding acquisition:** Alessio Alexiadis.

**Supervision:** Alessio Alexiadis.

**Validation:** Andrea Albano.

**Writing – original draft:** Andrea Albano.

**Writing – review & editing:** Alessio Alexiadis.

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
