## [Decision Letter · Decision Letter 0]

15 Jul 2020

PONE-D-20-07188

A Smoothed Particle Hydrodynamics study of Rayleigh collapse for a cylindrical cavity

PLOS ONE

Dear Dr. Albano,

Thank you for submitting your manuscript to PLOS ONE. After careful consideration, we feel that it has merit but does not fully meet PLOS ONE’s publication criteria as it currently stands. Therefore, we invite you to submit a revised version of the manuscript that addresses the points raised during the review process.

Please make revisions according to the reviewer's and editor's comments.

We look forward to receiving your revised manuscript.

Kind regards,

Michael H Peters

Academic Editor

PLOS ONE

Journal Requirements:

Additional Editor Comments (if provided):

This is a fundamental study on Rayleigh collapse of a cylindrical cavity using a computational technique called smoothed particle hydrodynamics. It is suitable for publication with minor revisions following the first referee's comments and my technical review comments below:

1. Since this study involves the use of computational software, it would need to be deposited as an open access software. I personally use GitHub, but please consult the PLoS One requirements on software for guidance.

2. The energy balance equation includes both viscous dissipation and thermal diffusion. It is not clear if these effects are being separated out when the authors talk about investigating "heat diffusion". The wording "heat diffusion" is awkward. Heat is the transfer of energy by virtue of temperature differences. We often use the term thermal diffusion when referring to Fourier's Law since it is expresses heat as being proportional to temperature differences. It is not clear how viscous dissipation is separated or lumped into the energy analysis.

3. There is a typo in Eq. 8, the pressure tensor is missing.

Reviewers' comments:

Reviewer's Responses to Questions

**Comments to the Author**

1. Is the manuscript technically sound, and do the data support the conclusions?

Reviewer #1: Yes

2. Has the statistical analysis been performed appropriately and rigorously? 

Reviewer #1: N/A

3. Have the authors made all data underlying the findings in their manuscript fully available?

Reviewer #1: Yes

4. Is the manuscript presented in an intelligible fashion and written in standard English?

Reviewer #1: Yes

5. Review Comments to the Author

Reviewer #1: Review on ‘A Smoothed Particle Hydrodynamics study of Rayleigh collapse for a cylindrical cavity’

Authors of this manuscript consider the collapse problem for an empty and gas-filled cavity. Authors propose a numerical approach which allows them to directly solve governing hydrodynamic equations for the cavity dynamics taking into account heat transfer. The main result of this manuscript is that the heat transfer may have a substantial impact on bubble’s collapse and should be taken into consideration when certain conditions are satisfied.

On the whole, I believe that this manuscript is suitable for publication. However, I have some remarks which authors should take into consideration:

1) Rayleigh collapse is usually associated with a spherical cavity. Thus, it seems that it might be more appropriate to change the title of the manuscript to ‘A Smoothed Particle Hydrodynamics study of the collapse for a cylindrical cavity’

1) It may be useful to compare the results of this manuscript with the results of the works where heat transfer was taken into account in Rayleigh-Plesset like models (see, e.g. A. Prosperetti, J. Fluid Mech. 222 (1991) 587–616., Stricker L., Prosperetti A., Lohse D. J. Acc. Soc. Am. 130 (2011) 3243-3251.)

3) For the verification of the numerical approach authors may also use the experimental and theoretical results from works D. Obreschkow, M. Bruderer, M. Farhat, Phys. Rev. E. 85 (2012) 066303, D. Obreschkow, M. Tinguely, N. Dorsaz, P. Kobel, A. de Bosset, M. Farhat, Exp. Fluids. 54 (2013) 1503.

4) Reference 15 has two authors. Thus, it should be cited in the text of the manuscript either by its number or by the last names of both authors. I would also like to draw attention of the authors to another work on analytical studying of bubbles dynamics (N.A. Kudryashov, D.I. Sinelshchikov, Phys. Lett. A 379 (2015) 798–802.), where surface tension was taken into account.

6. PLOS authors have the option to publish the peer review history of their article (what does this mean?). If published, this will include your full peer review and any attached files.

Reviewer #1: No

---

## [Author Response · Author response to Decision Letter 0]

3 Aug 2020

- Editor

 - Since this study involves the use of computational software, it would need to be deposited as an open access software. I personally use GitHub, but please consult the PLoS One requirements on software for guidance.

Answer:

Open access software was used for the simulation and data post process. All relevant information was added in the revised manuscript: see new section Software for simulation, visualisation and post-process in Model page 7, lines 204-207.

 - The energy balance equation includes both viscous dissipation and thermal diffusion. It is not clear if these effects are being separated out when the authors talk about investigating "heat diffusion". The wording "heat diffusion" is awkward. Heat is the transfer of energy by virtue of temperature differences. We often use the term thermal diffusion when referring to Fourier's Law since it is expresses heat as being proportional to temperature differences. It is not clear how viscous dissipation is separated or lumped into the energy analysis.

Answer:

In Eq. 9, the first term of the right hand side represents the sum of the reversible rate of internal energy increase by compression and the irreversible rate of internal energy increase by viscous dissipation. The second term is the particle form of the internal energy change by heat conduction following the Fourier’s law. In the adiabatic collapse simulation only the first term of Eq. 9 is accounted for. In the heat diffusive collapse, both term of Eq. 9 are used in the simulation.

In the revised manuscript, this was added at page 4 lines 100-104 and page 11 lines 324-326.

 - There is a typo in Eq. 8, the pressure tensor is missing.

Answer:

The typo has been corrected. Besides, Eq. 8 has been re-written to better mirror its particle form counter part. 

Additional note for the editor:

 During the revision of this manuscript Joshi et al published another work where they explain better how their model has been developed. For this page 2 lines 56-59 have been re-written and a new reference has been added.

- Reviewer 1

On the whole, I believe that this manuscript is suitable for publication. However, I have some remarks which authors should take into consideration:

- Rayleigh collapse is usually associated with a spherical cavity. Thus, it seems that it might be more appropriate to change the title of the manuscript to ‘A Smoothed Particle Hydrodynamics study of the collapse for a cylindrical cavity’"

Answer:

The title was changed as suggested.

- It may be useful to compare the results of this manuscript with the results of the works where heat transfer was taken into account in Rayleigh-Plesset like models (see, e.g. A. Prosperetti, J. Fluid Mech. 222 (1991) 587–616., Stricker L., Prosperetti A., Lohse D. J. Acc. Soc. Am. 130 (2011) 3243-3251.)

Answer:

We are aware of these publications. However, these works refer to oscillating bubbles, whereas our study focuses on the full bubble collapse. We prefer not to include them in the revised manuscript to avoid the risk of confusing the reader about the main focus of our paper.

- For the verification of the numerical approach authors may also use the experimental and theoretical results from works D. Obreschkow, M. Bruderer, M. Farhat, Phys. Rev. E. 85 (2012) 066303, D. Obreschkow, M. Tinguely, N. Dorsaz, P. Kobel, A. de Bosset, M. Farhat, Exp. Fluids. 54 (2013) 1503.

Answer:

These works are now quoted in the revised manuscript at page 10 lines 288-289 and page 11 lines 331-333.

- Reference 15 has two authors. Thus, it should be cited in the text of the manuscript either by its number or by the last names of both authors. I would also like to draw attention of the authors to another work on analytical studying of bubbles dynamics (N.A. Kudryashov, D.I. Sinelshchikov, Phys. Lett. A 379 (2015) 798–802.), where surface tension was taken into account..

Answer:

These works are now quoted in the revised manuscript at page 2 lines 29-32.

---

## [Decision Letter · Decision Letter 1]

15 Sep 2020

A Smoothed Particle Hydrodynamics study of the collapse for a cylindrical cavity

PONE-D-20-07188R1

Dear Dr. Albano,

We’re pleased to inform you that your manuscript has been judged scientifically suitable for publication and will be formally accepted for publication once it meets all outstanding technical requirements.

Kind regards,

Michael H Peters

Academic Editor

PLOS ONE

Additional Editor Comments (optional):

Reviewers' comments:

Reviewer's Responses to Questions

**Comments to the Author**

1. If the authors have adequately addressed your comments raised in a previous round of review and you feel that this manuscript is now acceptable for publication, you may indicate that here to bypass the “Comments to the Author” section, enter your conflict of interest statement in the “Confidential to Editor” section, and submit your "Accept" recommendation.

Reviewer #1: All comments have been addressed

2. Is the manuscript technically sound, and do the data support the conclusions?

Reviewer #1: Yes

3. Has the statistical analysis been performed appropriately and rigorously? 

Reviewer #1: I Don't Know

4. Have the authors made all data underlying the findings in their manuscript fully available?

Reviewer #1: (No Response)

5. Is the manuscript presented in an intelligible fashion and written in standard English?

Reviewer #1: Yes

6. Review Comments to the Author

Reviewer #1: In my opinion the authors revised the manuscript in accordance with all remarks and I believe that the manuscript can be published in the present form.

7. PLOS authors have the option to publish the peer review history of their article (what does this mean?). If published, this will include your full peer review and any attached files.

Reviewer #1: No

---

## [Editor Report · Acceptance letter]

18 Sep 2020

PONE-D-20-07188R1 

A Smoothed Particle Hydrodynamics study of the collapse for a cylindrical cavity  

Dear Dr. Albano:

I'm pleased to inform you that your manuscript has been deemed suitable for publication in PLOS ONE. Congratulations! Your manuscript is now with our production department. 

Kind regards, 

on behalf of

Michael H Peters 

Academic Editor

PLOS ONE